# Heterobimetallic Complexes of Bi- or Polydentate N-Heterocyclic Carbene Ligands and Their Catalytic Properties

Csilla Enikő Czégéni , Ferenc Joó * , Ágnes Kathó and Gábor Papp

Department of Physical Chemistry, University of Debrecen, P.O. Box 400, H-4002 Debrecen, Hungary;
nagy.csilla@science.unideb.hu (C.E.C.); katho.agnes@science.unideb.hu (Á.K.);
papp.gabor@science.unideb.hu (G.P.)
* Correspondence: joo.ferenc@science.unideb.hu

**Abstract:** This review summarizes developments in the synthesis and catalytic applications of those heterobimetallic carbene complexes in which at least two different metals are bound to the same ligand by at least one M-C(carbene) bond each. Several new synthetic methods for such complexes yielding well-defined and thoroughly characterized compounds are presented. The new complexes were found to be catalytically active in several (most often tandem) reactions. In certain cases, the incorporation of two different metals into the same imidazole- or triazol-based NHC-carbene complex resulted in the substantially higher catalytic activity of the heterobimetallic complex compared either to its homobimetallic analogs or to mixtures of comparable mononuclear complex fragments containing the two metals independently. This is a clear demonstration of advantageous metal–metal cooperation within the catalyst. Opposite examples are also discussed, where the heterobimetallic carbene complex proved inferior in relation to its homobimetallic analogs or to mixtures of homonuclear fragments.

**Keywords:** bidentate and polydentate NHC ligands; bridged di-NHC ligands; heterobimetallic carbene complexes; heterobimetallic catalyst; N-heterocyclic dicarbenes; 1,2,3-triazolyldiylidenes; 1,2,4-triazolyldiylidenes





## 1. Introduction

The art of the synthesis of metal-based heterogeneous catalysts is in finding the proper metal, additives, and support with the best match for the interaction of these components in order to obtain high activity, selectivity, and stability in a catalytic reaction. Catalysts containing two or more metallic components are widely applied, either as alloys supported on a seemingly inert solid carrier or as metals showing a strong interaction (SMSI) with a metal-containing support (e.g., suitable metal oxides [1]). The underlying interactions leading to the desired catalytic properties are scrutinized with much effort to find the perfect catalyst for a given process.

In contrast, soluble transition-metal-based catalysts applied in homogeneous solutions are, in most cases, monometallic complexes. This may originate from the fact that during the history of homogeneous catalysis by metal complexes in solution, sophisticated methods were developed for fine-tuning the electron density on the central metal ion and for shaping the three-dimensional structure of the complex as a whole to achieve the desired catalytic activity and selectivity. Obviously, what may be an optimal match for a certain metal ion and a set of ligands may not be optimal for a different metal ion with the same ligands. Furthermore, bimetallic complexes may not always show the hoped-for synergistic effect of the two metal ions. In addition, when two metal ions form a bimetallic complex, $M^1–D^1\sim D^2–M^2$, with a bidentate ligand, $D^1\sim D^2$, having two different donor sites ($D^1$ and $D^2$), a mixture of heterobimetallic and homobimetallic dinuclear complexes may form, or full disproportionation to $M^1–D^1\sim D^2–M^1$ and $M^2–D^1\sim D^2–M^2$ may even occur. In other

cases, the structure of the di- or polydentate ligand may be so rigid that it prevents the direct interaction between the coordinated metal ions. In certain reactions, however, the central metal ion in the catalyst complex may coordinate to another metal fragment as one of the ligands. This is the case for $[PtH(P_2)(SnCl_3^-)]$-catalyzed hydroformylation ($P_2$ = a bisphosphine or two monophosphines), where the direct interaction between the two metals plays an important role in determining the catalytic properties [2–4].

The idea of taking advantage of the interaction between two different metal ions in the same complex for better catalytic performance has a long history. Obvious candidates for such catalysts were mixed-metal clusters with direct metal–metal bonds. Although several examples of synergism in catalysis by such mixed-metal clusters were discovered, the widespread application of such catalysts was hampered by their instability, resulting in the formation of several catalytically active fragments [5]. The application of catalysts in which the two different metal ions were bridged by heterodifunctional ligands proved more successful [6–9], especially with early/late heterobimetallic complexes [10]. It was recognized from the beginning of such studies that in bimetallic dinuclear complexes, the two metal ions may not only cooperate in the catalysis of a given reaction but also act independently, allowing consecutive catalytic transformations. For example, Pittman and Smith achieved the cyclooligomerization of butadiene, followed by the hydrogenation of the resulting cycloalkenes to cycloalkanes with a catalyst obtained by anchoring both $[Ni(CO)_2(PPh_3)_2]$ and $[RhCl(PPh_3)_3]$ onto a phosphinated styrene–divinylbenzene resin (in other words, onto a solid macromolecular multidentate phosphine ligand) via phosphine exchange [11]. The same sequential reaction was also catalyzed by an assembly of $[Ni(CO)_2(PPh_3)_2]$ and $[RuCl_2(CO)_2(PPh_3)_2]$ anchored onto the same phosphinated polymer. The polymer-anchored catalysts could be efficiently recovered by filtration, and the Ni/Rh catalyst retained its catalytic activity for the repeated sequential oligomerization/hydrogenation of butadiene. In contrast, the heterogenized Ni/Ru catalyst lost its activity as early as in the first run due to the decomposition of the anchored $[Ni(CO)_2(PPh_3)_2]$ complex under the harsh conditions required for the activation of $[RuCl_2(CO)_2(PPh_3)_2]$ for the hydrogenation step [11]. This observation also demonstrated that for sequential reactions, the two metal-complex units (bridged by a multidentate ligand or fixed onto the same polymeric or solid support) must be sufficiently stable under the reaction conditions of both consecutive catalytic transformations.

For a long time, tertiary phosphine ligands played a central role in organometallic chemistry and catalysis. However, following the isolation of free N-heterocyclic carbenes [12], the introduction of N-heterocyclic carbene ligands (NHCs) [13] opened new possibilities for the synthesis and catalytic application of organotransition metal complexes [14,15]. Very often, NHCs form stronger bonds to metal ions than phosphines, which is an important requirement for catalyst stability. In addition, NHCs show unprecedented versatility in their composition, structure, and metal-binding properties. However, in comparison to tertiary phosphines, transition-metal complexes with N-heterocyclic ligands are still underrepresented in bimetallic catalysis.

In this short review, we survey the latest developments in the field with the aim of pointing out both the possibilities and difficulties of the synthesis of heterobimetallic NHC complexes and their use in catalysis. This work complements important earlier accounts [16–19]. In the selection of specific literature examples, we restricted our discussion to cases where at least two different transition-metal ions (M and M′) were bound in a complex with at least one metal–*C* (NHC carbene) bond each. The NHC ligands in these complexes were either monomeric compounds with at least two carbon atoms available for metal–*C* (NHC carbene) bond formation or dimeric (often bridged) or polymeric ligands with at least two different metal ions coordinated to carbene donor atoms in either chelate or bridging positions.

It is important to mention that in the studies cited in this review, the authors very thoroughly characterize all new compounds (first of all, the new heterobimetallic NHC complexes) with a plethora of relevant spectroscopic data, together with high-resolution

mass spectrometry. Importantly, in many cases, it was possible to isolate good-quality crystals for X-ray diffraction measurements, which provided suitable data for the unambiguous determination of the solid-state structures of the new ligands (precursors) and complexes. To avoid excessive repetition, structural features are rarely touched on in this review, and interested readers are advised to consult the original publications.

## 2. Synthesis of Heterobimetallic Complexes of Bi- or Polydentate NHC Ligands

### 2.1. General Considerations

In principle, any bidentate or polydentate NHC ligand may be suitable for binding two different metal ions. However, the inherent difficulty of the synthesis is caused by the often very similar (or identical) chemical environments and, hence, equivalent coordination properties of the carbenic carbon atoms, such as those shown by the bridged di-NHC complexes **1** and **2** (Scheme 1). Note that **2** is a so-called Janus-type NHC complex, in which the interaction between the two metal ions is hindered by the rigid link between the two 1,3-imidazole-2-ylidene units. The sequential metalation of the carbene precursors with different metal ions may be aided by the coordination of other donor atoms in the molecule. The dissimilar reactivities of the metal complexes used for the introduction of M and M′, such as in metalation via deprotonation of N-CH-N or in the oxidative addition of N-C(X)-N (X = halide) to a low-valent metal ion, may also offer routes for the synthesis of heterobimetallic NHC complexes. Several ingenious processes have been developed for such syntheses, examples of which are shown in the following sections.

**Scheme 1.** Examples of N-heterocyclic carbene ligands and their coordination possibilities in bimetallic complexes.

Before going into details, it is instructive to consider the coordination possibilities of N-heterocyclic carbenes. Upon deprotonation, 1,3-dialkylimidazolium salts (exemplified in Scheme 1 by 1,3-dimethylimidazolium chloride, **3**) usually yield 1,3-dialkylimidazole-2-ylidenes (normal NHCs), which may coordinate to a metal ion through their C2 carbene atom (**4**, Scheme 1). In contrast, metal coordination on C4 (abnormal NHCs, **5**, first observed by Crabtree and co-workers [20]) (Scheme 1) also happens frequently. (Since a free abnormal carbene cannot be described by a reasonable resonance form without additional charge, such compounds are often called mesoionic carbene—MIC—ligands.) The difference in the coordination properties of C2 and C4 may lead to the formation of heterobimetallic complexes (**6**, Scheme 1). In contrast, dicationic triazolium, e.g., 1,2,4-trimethyltriazolium salts, can be used as versatile precursors to ditopic carbene ligands and their complexes, exemplified by **7** (Scheme 1). Similarly, 1,2,3-triazoles (readily available through alkyne/azide cycloaddition reactions) afford various mesoionic carbene ligands, such as **8** (Scheme 1), for the synthesis of both mono- and heterobimetallic NHC complexes [18]. Similar to the case of imidazole-ylidenes, the precursors to triazole-derived carbenes can also be connected by suitable linker units, giving rise either to free polydentate (or even polymeric) carbene ligands or directly to their metal complexes.

### 2.2. Synthesis and Catalytic Application of Heterobimetallic NHC Complexes Containing Imidazolium-Derived Carbene Ligands

### 2.2.1. Unassisted Syntheses of Heterobimetallic NHC Complexes and Their Catalytic Properties

In this section, we review the synthetic methods of those NHC complexes in which—in most cases—the formation of the heterobimetallic compounds was not assisted by other interactions (e.g., cyclometalation) or coordination to other donor groups (anchors) of the ligand.

The simplest way to obtain a bis(imidazole-ylidene) carbene ligand precursor is to link two imidazolium units by a suitable alkyl, aryl, or other kind of bridge between the two unsubstituted nitrogens [21,22]. This technique was used by Cowie and co-workers for the synthesis of **11** (Scheme 2) [23]. One of the C2 carbenes was coordinated to Pd, while the remaining imidazolium moiety was metalated by [Rh($\mu$-OAc)(cod)]$_2$ to yield the Pd/Rh heterobimetallic NHC complex **11** (the acetate served as an internal base). The cod ligand (1,5-cyclooctadiene) could be easily replaced by CO, affording the dicarbonyl product. The analogous Pd/Ir(cod), Pd/Ir(CO)$_2$, and Ir/Rh complexes were also prepared and characterized. However, no catalysis by these heterobimetallic NHC complexes was reported.

**Scheme 2.** Synthesis of the Pd/Rh heterobimetallic complex **10** containing a methylene-linked diimidazole-diylidene ligand.

Braunstein and co-workers reported the synthesis of the bis(N-ethylimidazolium) compound **12**, where the two imidazolium units were attached to an *m*-xylyl linker (Scheme 3) [24]. The deprotonation of **12** under the effect of LiN(SiMe$_3$)$_2$ afforded the free dicarbene. Interestingly, the reaction of this dicarbene with [Ir($\mu$-Cl)(cod)]$_2$ did not lead to the formation of a chelate Ir complex; instead, the reaction gave either the classical mono- or bis-carbene Ir-NHC complexes. This observation opened the way for sequential metalation, in which the free dicarbene was first reacted with [Ir($\mu$-Cl)(cod)]$_2$ with the formation of **12**, followed by another metalation step using [Rh($\mu$-Cl)(cod)]$_2$, which led to the heterobimetallic product **13**. The new NHC complexes were not applied as catalysts.

**Scheme 3.** Synthesis of the Ir/Rh heterobimetallic complex **13** containing a diimidazole-diylidene ligand with a m-xylyl linker.

Teng and Vinh Huyn prepared a bis(carbene) precursor that contained an N,N,N-pincer linker between two alkylbenzimidazolium moieties [25]. Capitalizing on the 'borderline' hard/soft coordination properties of Pd(II), they synthesized **14**, in which the palladium ion is coordinated only by hard donor atoms (Scheme 4). In reaction with a half equivalent of Ag₂O, chloride abstraction from Pd led to **15**. In addition to the three pincer N donor atoms, in this complex, Pd is also bound to a benzimidazole-ylidene unit, while the molecule still contains a non-metalated benzimidazolium part. The metalation of the latter with the use of Ag₂O and [AuCl(THT)] (THT = tetrahydrothiophene) afforded the heterobimetallic Pd/Au complex **16**. The analogous Pd/Pd complex was also prepared. Although the potential use of this ligand for the preparation of cooperative heterobimetallic bis(NHC)-complex catalysts was mentioned, no such experiments were described in the paper.

**Scheme 4.** Synthesis of the Pd/Au heterobimetallic complex **16** based on the borderline soft/hard coordination ability of Palladium.

A mixed imidazolium/benzimidazolium bis(carbene) precursor, 17 (Scheme 5), was synthesized by Hahn, Peris, and co-workers [26]. The reaction of this compound with 1 equiv [IrCl₂(Cp*)]₂ (Cp* = pentamethylcyclopentadienyl anion) in the presence of NaOAc as a base led to the formation of a mixture of monometallic iridium complexes. However, under similar conditions, [RhCl₂(Cp*)]₂ selectively yielded **18**, in which Rh binds to the C2 carbenic carbon of the imidazole moiety and simultaneously orthometalates the phenyl ring of the benzimidazolium unit (Scheme 5). The formation of such a metallacycle contributes to the selective formation of the imidazol-2-ylidene type complex, while the benzimidazolium part remains available for further reactions. Indeed, the benzimidazole-ylidene complexes Rh/Pd (**19**), Rh/Ir (**20**), Rh/Au (**21**), and Rh/Ru (**22**) were prepared from **18** with the use of the Ag₂O-based method of synthesis. For a comparison of catalytic purposes, several other heterobimetallic complexes and their monometallic fragments (**23–28**) were synthesized (Scheme 5).

The catalytic properties of the new heterobimetallic NHC complexes were examined in two reactions. First, the tandem Suzuki–Miyaura *C–C* coupling/transfer hydrogenation of 4-bromoacetophenone was studied with phenylboronic acid as a coupling partner and 2-propanol as a hydrogen donor and solvent (Scheme 6). At a temperature of 100 °C, and with Cs₂CO₃ as a base, the catalytic reactions proceeded with 99% conversion and afforded only the coupled product (4-phenylacetophenone) and its hydrogenated derivative 1-(4-biphenyl)ethanol. The reaction in the absence of a metal catalyst yielded 54% 4-Br-phenylethanol, but no debromination of the latter to 1-phenylethanol was observed. Concerning the relative activities of heterobimetallic NHC complexes, **19** (Rh/Pd) actively catalyzed the *C–C* coupling of 4-bromoacetophenone with phenylboronic acid, but its activity in the further transfer hydrogenation of the resulting biphenyl-methylketone was lower than that of **23** and **24**. The latter two catalysts provided 82% and 84% yields of 1-(4-biphenyl)ethanol, respectively, in contrast to **19**, which afforded only 53% (all yields at 99% total conversion of 4-bromoacetophenone). On the basis of kinetic measurements, the reaction could be regarded as a conventional consecutive reaction in which the first interme-

diate, the product of *C–C* coupling, accumulated up to 85% before further hydrogenation by H-transfer from 2-propanol. This behavior shows that the Pd-catalyzed *C–C* coupling proceeds faster than the Rh-catalyzed transfer hydrogenation. Nevertheless, cooperation between the two metals in the heterobimetallic complexes cannot be excluded since, in certain cases, the reactions proceeded faster with heterobimetallic catalysts, e.g., **23** and **24**, than with equimolar mixtures of the monomeric 'fragments', such as (**25** + **28**), and (**26** + **28**).

**Scheme 5.** Synthesis of various heterobimetallic complexes **19**–**24** and their analogous monometallic molecular fragments prepared for comparison of catalytic activities.

With the same catalyst, 4-bromoacetophenone was coupled to phenylboronic acid in *n*-butyl alcohol, in which case, the product mixture contained α-*n*-butylated products, too (Scheme 6) [26]. Catalysts **23** and **24** were found more effective for α-*n*-butylation than mixtures of (**25** + **27**) and (**26** + **27**). Again, the time-dependent reaction profile showed a typical consecutive reaction. The higher activity of heterobimetallic complexes relative to the monomeric analogs suggested some additivity or cooperativity between the two metal ions in the catalyst complexes; however, the mechanism of such cooperations was not established.

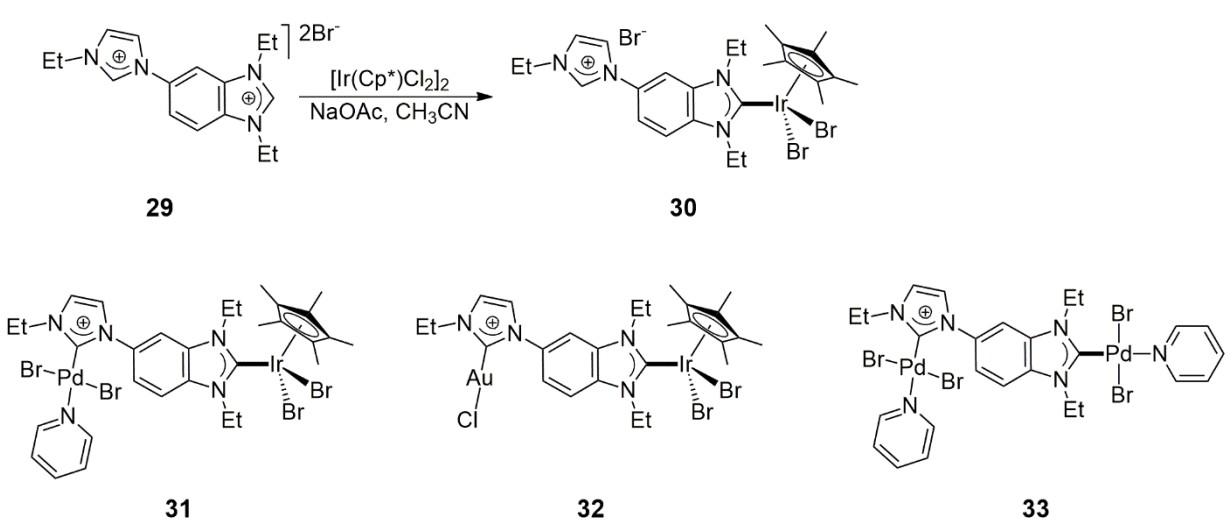

**Scheme 6.** Tandem Suzuki-Miyaura C-C coupling/transfer hydrogenation of 4-bromoacetophenone in 2-propanol (**a**) and in *n*-butanol (**b**).

In a related study, Nishad, Kumar, and Rit [27] found that reaction of **29** and 0.5 equiv [IrCl$_2$(Cp*)]$_2$ afforded selectively the monometallic Ir-carbene complex **30**, with a pendent N-ethylimidazolium unit. Further metalations with Pd(OAc)$_2$ or Ag$_2$O/Au(THT) yielded the bimetallic complexes **31** (Ir/Pd) and **32** (Ir/Au), respectively. For comparison, the analogous Pd/Pd homobimetallic complex **33** was also prepared (Scheme 7).

**Scheme 7.** Synthesis of various heterobimetallic complexes **30**–**33**, capitalizing on the different reactivities of the two pro-carbene C atoms in the imidazolium and benzimidazolium parts of **29**.

Complexes **31** (Ir/Pd) and **33** (Pd/Pd) were used as catalysts in the tandem Suzuki–Miyaura coupling of 4-bromoacetophenone with phenylboronic acid, followed by transfer hydrogenation with the use of 2-propanol and KO$^t$Bu (O$^t$Bu = tert-butylate) (Scheme 6a). Under optimal conditions (1 mol% catalyst, 100 °C, 12 h, 2-propanol as solvent) the Ir/Pd catalyst (**31**) led to the formation of the *C*–*C*-coupled and hydrogenated product with an 87% yield. Under the same conditions, catalysis by **33** (Pd/Pd) yielded only 61% 1-(4-biphenyl)ethanol, accompanied by 27% 1-phenylethanol arising from the debromination

and transfer hydrogenation of 4-bromoacetophenone. These results suggested that there was some cooperation between the metals in the heterobimetallic catalyst, leading to higher activity in the case of the Ir/Pd catalyst over the Pd/Pd complex. Interestingly, under similar experimental conditions to the above, **31** also showed high catalytic activity in the tandem defluorination/transfer hydrogenation of mixed (4-fluorophenyl)-alkyl- or arylketones, in which it led to the formation of the defluorinated products in 80%, 97%, and 96% (alkyl = methyl, ethyl; aryl = phenyl, respectively). In these reactions, too, 2-propanol served both as a hydrogen donor and solvent.

Hahn, Peris, and co-workers applied the 1,4-phenylene-bridged (**34**) and the 1,3-phenylene-bridged (**35**) N-methylimidazolium iodides to the synthesis of heterobimetallic Rh/Au (**36**, **37**) and Ir/Au (**38**, **39**) complexes (Scheme 8) [28]. In the first step of sequential metalation, **34** and **35** reacted with either [RhCl$_2$(Cp*)]$_2$ or [IrCl$_2$(Cp*)]$_2$, yielding the corresponding cyclometalated Rh/NHC or Ir/NHC intermediates, respectively. Next, these intermediates were treated with Ag$_2$O/Au(THT), affording the desired heterobimetallic Rh/Au (**36**, **38**) and Ir/Au (**37**, **39**) products. Overall yields were in the 21–56% range.

**Scheme 8.** Cyclometalation-assisted synthesis of the Rh/Au (**36**, **37**) and Ir/Au (**38**, **39**) heterobimetallic complexes containing 1,4-phenylene or 1,3-phenylene bridges between the two imidazolium units in **34** and **35**.

The catalysts were tested in the tandem condensation of nitrobenzene with benzyl alcohol and imine reduction (Scheme 8) with a 0.5% catalyst loading and Cs$_2$CO$_3$ as a base at 100 °C for 22 h in benzyl alcohol as the solvent. With the exception of **38**, all other new heterobimetallic NHC-complex catalysts led to 100% conversion, which was higher than the conversions generally obtained with equimolar mixtures of analogous monometallic 'fragments'. The highest imine yield (62%) was obtained with catalyst **36** (Rh/Au) with only 13% secondary amine (the hydrogenated product). It could be concluded that the cooperation of the two metal ions in the heterobimetallic complexes with *meta*-bis-NHC ligands was stronger than in those with *para*-bis-NHC ligands, which was demonstrated by the highest activity of **39** for amine formation (82%).

Gonell, Poyatos, Mata, and Peris synthesized a Y-shaped tris-N-heterocyclic carbene ligand precursor and applied it to the synthesis of Rh-, Ir-, and Pd-based homo- and heterobimetallic complexes **40**, **41**, **42**, and **43** by sequential metalation with the use of [RhCl(cod)]$_2$, [IrCl(cod)]$_2$, or Pd(OAc)$_2$ (Scheme 9) [29]. It was found that this Y-shaped ligand reacted with the first equivalent of the metal compounds with the formation of chelate bis-NHC complexes, which—upon the addition of the next equivalent of metal compounds—could bind another metal ion on the C2 carbene atom of their third imidazolium group. The order of addition of the metal precursor to the ligand determined which

metal ion was coordinated in a chelate or monodentate fashion (see **40** vs. **41**). Interestingly, despite the use of [RhCl(cod)]$_2$ for the preparation of **43** in the presence of NaOAc, the chelated rhodium was always in the Rh(III) oxidation state. The solid-state structures of **40** and **43** were determined by X-ray diffraction, which showed a through-space Ir-Pd distance of 6.76 Å in **40** and a Rh(I)-Rh(III) distance of 7.003 Å in **43**. For comparison purposes, the homobimetallic compound **44** (Pd/Pd) was also prepared.

**Scheme 9.** Synthesis of the Pd/Ir (**40**, **41**) and Ir/Rh (**42**, **43**) heterobimetallic complexes via sequential metalation of a Y-shaped tris-N-hetrocyclic carbene ligand precursor.

In the tandem dehalogenation/transfer hydrogenation of 4-bromoacetophenone (Scheme 10), the Ir(chelate)/Pd(monodentate) catalyst **41** yielded 1-phenylethanol in close to quantitative yield (1–2% catalyst loading, 100 °C, 20–48 h, 2-propanol as a solvent and H-donor). In contrast, Ir(monodentate)/Pd(chelate) **40** was much less active in transfer hydrogenation and, under the same conditions, led to the formation of acetophenone in 52–89% yields. A 1:1 mixture of related (albeit not strictly analogous) binuclear Ir/Ir and Pd/Pd complexes catalyzed the dehalogenation step efficiently; however, the transfer hydrogenation proceeded only to 25%. As expected, **41** was also an effective catalyst for the tandem Suzuki–Miyaura coupling/transfer hydrogenation of 4-bromoacetophenone with phenylboronic acid (Scheme 6), in which the ratio of the biphenyl methyl ketone and 1-(4-biphenyl)ethanol products strongly varied with the reaction time (from 82% to 16% and from 0% to 69%, respectively, in the 0.5–20 h range of reaction time). It could be concluded that there was substantial catalytic cooperation between the two metal ions; still, Pd was mainly responsible for dehalogenation, while Ir was necessary for the effective catalysis of transfer hydrogenation.

**Scheme 10.** Transfer hydrogenation of 4-bromoacetophenone in 2-propanol catalyzed by **40** and **41**.

It was also demonstrated that **40** and **41** were very active catalysts for the cyclization of 2-(*ortho*-)aminophenyl ethyl alcohol to indole (GC yields > 99%) in toluene at 110 °C for 2 h. Similarly, the same complexes catalyzed the acylation of bromobenzene with *n*-hexanal,

resulting in its full conversion to butyl phenyl ketone at 115 °C in DMF for 16 h. However, the details of these reactions were not reported.

Hemmert, Gornitzka, and co-workers were interested in the effect of a Au-NHC unit on the photophysical properties of $[Ru(bipy)_3]^{2+}$ (bipy = 2,2′-bipyridine) within the same heterobimetallic complex [30]. Conversely, they also wished to reveal the effect of the $[Ru(bipy)_3]^{2+}$ part of the molecule on the cytotoxic, antileishmanial, and antimalarial activities of the Au-NHC moiety. For this purpose, a 2,2′-bipyridine unit was attached via a methylene linker to methyl-, *n*-butyl-, and benzylimidazole in the reaction of the azole with 5-(bromomethyl)-2,2′-bipyridine. In the reaction of $[RuCl_2(bipy)_2]$ with the functionalized azolium salt, an octahedral Ru(II) complex was formed with three bipy ligands, one of which carried a pendant azolium group. With the treatment of this intermediate with $Ag_2O$ and then with $AuCl(SMe_2)$, heterobimetallic Ru-bipy/Au-NHC complexes **45** and **46** were obtained, differing only in the substituent of the NHC-carbene part (Scheme 11). The main conclusion of this study was that the photophysical properties of the Au/Ru heterobimetallic complexes were very similar to those of $[Ru(bipy)_3]^{2+}$ and that without Ru(II) coordination, the compounds showed low biological activity in comparison to the Au(I) complex. In line with the original purpose of the studies, no catalysis by compounds **45** and **46** was reported.

**Scheme 11.** Synthesis of the heterobimetallic complexes **45** and **46** used in photophysical and in biological studies.

In a related study [31], Gonell, Poyatos, and Peris synthesized an Au/Pt heterobimetallic NHC complex based on the Y-shaped tris-NHC ligand precursor developed by them earlier [29]. In the first step, the ligand precursor tris-azolium salt was reacted with the bis(alkynyl) complex $[Pt(CCPh_2)(cod)]$. This reaction resulted in complex **47**, which contained the $Pt(CCPh)_2$ moiety in chelate coordination. The reaction of this compound with a pincer-type (CNC)Au(I)Cl complex in the presence of NaOAc in acetonitrile led to the desired heterobimetallic Au/Pt product **48** (Scheme 12). In addition to NMR spectroscopy and HR mass spectrometry, **48** was characterized by UV-Vis absorption and emission spectroscopy, as well as by cyclic voltammetry and differential pulse voltammetry. Both **47** and **48** were found emissive in the solid state but non-emissive in solution. It could also be concluded that the emission was centered on the Pt(II) fragment.

Barnard and co-workers planned to prepare heterobimetallic complexes based on the pro-ligand **49** [32]. However, treatment of this ligand with 0.5 equiv of AuCl(THT) in the presence of a weak base such as NaOAc produced only a homobimetallic Au(I) complex, together with unreacted **49** and other unidentified products. A postmetalation ligand synthesis was developed, according to which the second imidazolium moiety was synthesized only after the initial metalation step with AuCl(THT). The ethylation of **50** with $Et_3OBF_4$ in $CH_3CN$ led to complex **51**, in which the second imidazolium was metalated to yield the corresponding Au/M complexes (**52**), where M = Cu$^+$, Ag$^+$, or Hg$^{2+}$ (Scheme 13). Although the complexes **52** were emissive in the solid state, they were not luminescent in solutions. Nevertheless, according to the solid-state structures (X-ray), all heterobimetallic

complexes support heterometallophilic interactions, and their structures may give further insight into the luminescent properties of such heterobimetallic compounds.

**Scheme 12.** Synthesis of the Pt/Au heterobimetallic NHC complex **48** used in photophysical studies.

**Scheme 13.** Synthesis of the heterobimetallic NHC complex series **51** (M = $Cu^+$, $Ag^+$ or $Hg^{2+}$) used for studies of luminescent properties of the complexes.

A different postmetalation synthetic method was developed by Hahn, Jin, and co-workers for the preparation of multidentate NHC complexes [33]. The Ir- or Rh-containing bis-NHC complexes **53** and **54**, which contained a protic NHC ligand unit, were reacted with trans-1,4-dibromo-2-butene or with 1,4-bis(dibromomethyl)benzene, leading to the N-alkylated (benzylated) derivatives with pendant bromomethyl groups. The further reaction of these intermediates with **53** or **54** gave tetrakis-NHC complexes, of which **55** is an Ir/Rh heterobimetallic NHC compound (Scheme 14). In addition to characterization by NMR and HRMS, the solid-state structure of **55** was determined by X-ray crystallography. However, no catalysis was reported for this new heterobimetallic complex. When **53** reacted with 1,3,5-tris(bromomethyl)benzene instead of 1,4-bis(bromomethyl)benzene, the same synthetic strategy led to the homometallic hexakis-NHC iridium complex **56** (Scheme 14).

Hahn, Li, and co-workers reported a general strategy for the synthesis of oligomeric or polymeric NHC complexes having strictly pre-determined positions of the NHC units and metal ions [34]. The strategy is based on the click reaction of suitable monomeric NHC–metal complexes (secondary building units or SBUs) having *p*-azidophenyl and *p*-ethynylphenyl NHC wingtips (Scheme 15). The azide group could be directly attached to the aromatic ring or via a flexible methylene linker. The reaction of the azide-containing SBUs with phenylethynyl-substituted NHC complexes resulted in the formation of polymers with alternating metal–NHC units joined together by 1,2,3-triazole (or the more flexible methylene-triazole) linkers along the polymeric chain. In cases when one of the SBUs was an azide-modified copper complex, the click reaction proceeded rapidly without an external catalyst and, depending on the combination and molar ratios of the SBUs, resulted in heterodimetallic dimers, trimers, or polymers (only the Cu/Ag bis-NHC complex **57** and the −[Cu~Ag]$_{n−}$ polymer **58** are shown in Scheme 15). The Cu/Ag derivatives could be further transmetalated in reaction with, e.g., allylpalladium(II) chloride dimer [PdCl(allyl)]$_2$ to yield analogous Cu/Pd polymers.

**Scheme 14.** Synthesis of the Ir/Rh heterobimetallic complex **55**.

**Scheme 15.** General method for synthesis of dimeric (**57**), oligo- or polymeric (**58**) heterobimetallic NHC complexes based on the alkyne-azide click reaction.

The Cu/Ag heterobimetallic bis-NHC complex **57** and the corresponding polymer **58** were studied in the catalytic alkynylation of 2,2,2-trifluoro-1-phenylethanone with various terminal alkynes (Scheme 16). Detailed measurements with phenylacetylene as a model substrate showed that both **57** and **58** exhibited high catalytic activity, leading to 96 and 95% yields of the fluorinated propargylic alcohol at a catalyst loading as low as 0.5 mol%. (For solubility reasons, **57** was used in THF, while **58** was in DMF.) Control experiments with a Cu/Ag polymer with a rigid triazol linker showed diminished catalytic activity (67% yield) compared to that of **58** (99% yield), pointing to the importance of the flexibility of the polymer chain. Importantly, a 1:1 mixture of [CuCl(SI*i*Pr)] (SI*i*Pr = 1,3-bis(2,6-isopropylphenyl)imidazoline-2-ylidene) and [AgCl(SI*i*Pr)] (mononuclear analogs of the NHC complex units in **57** and **58**) also showed much inferior activity (65% in THF and 39% in DMF). The latter observation strongly supports the assumption that there is cooperativity between Cu(I) and Ag(I) in the catalysis of this reaction, which also

requires the flexibility of the polymer. Other terminal alkynes also reacted smoothly with **57** and **58** as the catalysts.

**Scheme 16.** Alkynylation of 2,2,2-trifluoro-1-phenylethanone with phenylacetylene catalyzed by **57** and **58**.

　　　Anionic N-heterocyclic dicarbenes are mostly used as bridging ligands between two main-group elements. In a recent paper, Hevia et al. described the magnesiation of the common NHC-carbene IPr (1,3-bis(2,6-isopropylphenyl)imidazole-2-ylidene) [35]. The reaction of IPr and NaR (R = $CH_2SiMe_3$) in hexane yielded a polymeric anionic sodium–NHC intermediate, which could be further metalated with the use of $MgR_2$ in THF to yield the heterobimetallic product **59**. In this complex, magnesium was coordinated to the abnormal C4 carbene position of IPr (Scheme 17). The reaction of **59** and $AuCl(SMe_2)$ in toluene at −70 °C resulted in the novel Mg/Au heterobimetallic complex **60**. In this compound, magnesium retained its C4 coordination; however, one of the $CH_2SiMe_3$ groups was transferred from Mg to Au, while the coordination sphere of Mg was filled with an additional THF molecule originally bound to Na in **59**. The reaction is accompanied by the formation of NaCl, the toluene-insolubility of which also drives the process toward the formation of **60** by creating a coordination vacancy at C2 (Scheme 17). This study opened new pathways for the synthesis of anionic and abnormal Mg-NHC complexes and their application in transmetalation reactions, leading to new heterobimetallic Mg/Au complexes. No catalysis by the new compounds was reported.

**Scheme 17.** Synthesis of the Mg/Na (**59**) and Mg/Au (**60**) heterobimetallic NHC complexes featuring normal and abnormal coordination of the two metals to the same imidazolium diylidene ligand.

2.2.2. Cyclometalation-Assisted Synthesis of Heterobimetallic NHC Complexes and Their Catalytic Properties

　　　In addition to the formation of NHC complexes through metal–carbene coordination, product stability is often increased by the orthometalation of a suitable aryl substituent attached to an imidazolium nitrogen. Such a case was revealed by Maity, Hahn, and co-workers [36] in the reaction of 1,4-bis(3-ethyl-1-imidazolium)benzene **61**, originally synthesized by You et al. [37] and later by the Hahn group via a modified procedure [38]. As shown in Scheme 18, the reaction of **61** with $[IrCl_2(Cp^*)]_2$ yielded **62**, which afforded the heterobimetallic **63** upon further metalation with $[RhCl_2(Cp^*)]_2$ (both reactions took place in acetonitrile solution in the presence of $Cs_2CO_3$ and NaOAc). Both Ir(III) and Rh(III) cyclometalated the 1,4-phenylene linker between the NHC units. Orthometalation in **62** and double orthometalation in **63** were established by 2D NMR spectroscopy. Furthermore, the determination of the solid-state structure of **62** unambiguously confirmed this structural feature. The catalytic properties of **62** and **63** were not investigated.

**Scheme 18.** Cyclometalation-assisted formation of the Ir/Rh complex **63** with a 1,4-phenylene-bridged di-NHC ligand.

In a related study, Maity and co-workers synthesized 1,4-bis(3-methyl-1-imidazolium)-2,3-dimethyl-benzene **64** (Scheme 19) [39]. Although **64** is very similar to the previously mentioned ligand **61**, the two methyl substituents on the 1,4-phenylene linker prevented the double orthometalation of the bridging phenylene moiety. In contrast, the reaction of **64** with $[IrCl_2(Cp^*)]_2$ in acetonitrile afforded a monometallic cyclometalated Ir(III) NHC complex, which could be further metalated on the second NHC carbene with $PdCl_2$ in pyridine. The overall reaction yielded the PEPPSI-type Ir/Pd bimetallic bis-NHC complex **65**. An X-ray diffraction study of **65** revealed a 6.14 Å distance between the two metal centers in the solid-state structure. The similar reaction of **64** with $PdCl_2$/pyridine led to the formation of the analogous Pd/Pd complex **66** (no cyclometalation in this case).

**Scheme 19.** Cyclometalation-assisted synthesis of the Pd/Ir complex **65** prepared with the use of the carbene ligand precursor **64**.

The catalytic activity of the Ir/Pd heterobimetallic bis-NHC complex **65** was studied in the α-arylation of oxindole in toluene solutions (Scheme 20). In comparison to the homometallic counterpart **66**, the heterobimetallic complex **65** showed slightly higher activities both with bromobenzene (32% vs. 26% isolated yields) and with 2-bromotoluene (53% vs. 41% isolated yields). Similarly, in the Suzuki–Miyaura cross-coupling of 4-bromobenzaldehyde and phenylboronic acid, **65** and **66** showed comparable activities (99% vs. 83% conversions in 5 h). Under the same conditions, with 4-toluylboronic acid, the yield of the coupled product was 99% with both catalysts. Note that these reactions were carried out with water as a solvent. Conversely, when the solvent was changed for 2-propanol/THF and $KO^tBu$ was added as a base, tandem Suzuki–Miyaura coupling/transfer hydrogenation took place both with 4-bromobenzaldehyde and 4-bromoacetophenone as substrates. In these reactions, **65** was consistently more efficient, with yields up to 85%, than a 1:1 mixture of its mononuclear Pd(II) and Ir(III) counterparts (highest yield of 41%). It

was concluded that the outstanding activity of the Ir/Pd heterobimetallic bis-NHC complex in the tandem *C–C* cross-coupling/transfer hydrogenation reaction was most probably due to the interaction between the Ir and Pd centers in the molecule.

**Scheme 20.** α-Arylation of oxindole with bromoarenes catalyzed by **65** and **66**.

In continuation of their work described in [39], following the same synthetic strategy, Majumder, Maity, et al. applied ligand **64** to the synthesis of the Rh/Pd analog **67** of the Ir/Pd complex **65** [40]. Similar to **65**, **67** also showed orthometalation (this time with the formation of a rhodacycle), contributing to the complex's stability (Scheme 21). Analogous complexes with 4-methoxypyridine and 4-chloropyridine were also obtained and thoroughly characterized by NMR and HRMS methods and, in several cases, by X-ray diffraction measurements. The catalysis of the tandem Suzuki–Miyaura cross-coupling/transfer hydrogenation of 4-bromobenzaldehyde with phenylboronic acid in 2-propanol/THF with KO$^t$Bu as a base was attempted (Scheme 6). With the Rh/Pd heterobimetallic bis-NHC complex, **67** showed only negligible activity (6% isolated yield of 1-(4-biphenyl)ethanol at 90 °C, 2 h).

**Scheme 21.** Synthesis of the heterobimetallc Pd/Rh complex **64**.

The heterobimetallic tris-NHC complexes **68** (Rh/Pd) and **69** (IrPd) were obtained by the Hahn group with the selective stepwise metalation of the non-symmetric 1,2,4-tris(imidazolium)-substituted tris-NHC ligand precursor **70** [41]. In the reaction of **70** and Pd(OAc)$_2$ in DMF, a mononuclear chelate bis-NHC compound was formed, which could be further metalated (CH$_3$CN, Cs$_2$CO$_3$) with the use of [RhCl$_2$(Cp*)]$_2$ or [IrCl$_2$(Cp*)]$_2$ to yield **68** or **69**, respectively. It was also discovered that the same heterobimetallic complexes could be obtained in a one-step reaction of **70** by employing Pd(OAc)$_2$ together with [RhCl$_2$(Cp*)]$_2$ or [IrCl$_2$(Cp*)]$_2$, this time in the presence of NaOAc instead of Cs$_2$CO$_3$ (Scheme 22). Pd was always coordinated in a chelate fashion, while Rh or Ir was coordinated by a single NHC donor and orthometalated the phenyl ring. Thorough NMR and HRMS characterizations were supplemented by X-ray diffraction measurements, which showed an Ir-Pd through-space distance of 7.240 Å in **69**. Although this metal–metal distance is rather large for cooperative catalysis, this work exemplifies the simple access to heterobimetallic complexes, which may be useful in organic transformations. However, no attempt was made to reveal the catalytic properties of **68** and **69**.

**Scheme 22.** Cyclometalation-assisted formation of (**a**) Pd/Rh (**68**), Pd/Ir (**69**) heterobimetallic, and (**b**) Ir/Rh$_2$ (**73**), and Au/Rh$_2$ (**74**) heterotrimetallic NHC complexes from tris-imidazolium carbene precursors.

Maity, Schulte to Brinke, and Hahn investigated the complexation properties of a symmetric tris-NHC ligand, which was obtained from the 1,3,5-trisimidazolium salt **71** (Scheme 22) [42]. The reaction of this precursor with [RhCl$_2$(Cp*)]$_2$ in MeCN at 65 °C in the presence of Cs$_2$CO$_3$ and NaBr resulted in complex **72**, which contained two rhodacycles and one uncomplexed imidazolium unit. The further metalation of **72** was achieved with [IrCl$_2$(Cp*)]$_2$ or [AuCl(SMe)$_2$] (in both cases via the Ag$_2$O route) to yield Rh/Ir (**73**) or Rh/Au (**74**) heterobimetallic tris-NHC complexes, respectively. It is of interest that **73** is a triply orthometalated heterotrimetallic complex with three five-membered metalacycles fused to the central phenyl ring. From detailed NMR measurements, it was concluded that the metalation of the third imidazolium moiety did not affect the other two metals and that all metals behaved as independent electronic centers. No catalytic studies with the new heterobimetallic NHC complexes were reported.

### 2.2.3. Oxidative Addition Reactions in the Synthesis of Heterobimetallic NHC Complexes

The oxidative addition of a carbon–halogen bond of a suitable halogen-substituted N-heterocyclic compound onto a low-valent metal complex may yield N-heterocyclic carbene complexes. This possibility was extensively studied by the Hahn research group with the aim of developing versatile methods for the synthesis of heterobimetallic NHC complexes. Accordingly, the research described below did not include catalytic applications of the heterobimetallic products.

The *m*-xylyl-bridged imidazolium/chlorobenzimidazole ligand precursor **75** gave the imidazolylidene–Au complex **76**, in which the chlorobenzimidazole moiety remained available for metalation by oxidative addition to [Pd(PPh₃)₄] [43]. The Pd/Au heterobimetallic product **77** could be isolated both as the chloride (29%) and as the PF₆ salt (56%). An alternative pathway for the synthesis of **77** included the formation of the gold complex of imidazolium-2-ylidene in the first step, followed by the oxidative addition of the C-Cl bond to [Pd(PPh₃)₄]. This example clearly shows the possibility of sequential site-selective metalation due to the different reactivities of the N-CH-N and N-C(Cl)-N groups (Scheme 23).

**Scheme 23.** Synthesis of the Au/Pd₂ heterotrimetallic complex **77** via oxidative addition of C-Cl across Pd(0) in [Pd(PPh₃)₄].

Regioselectivity in double oxidative addition reactions of bis-NHC precursors also resulted in heterobimetallic bis-NHC complexes [44]. In the reaction of the 4,4′-*p*-biphenylene-bridged chlorobenzimidazole/ethylchlorobenzimidazolium compound **78** with one equivalent of [Pd(PPh₃)₄], the monopalladium NHC complex **79** was formed selectively in the oxidative addition reaction of the 1,3-dialkyl-2-chlorobenzimidazolium moiety (3 h, 75 °C, THF). In a slower reaction (18 h, 100 °C), [Pt(PPh₃)₄] underwent oxidative addition at the non-alkylated benzimidazole part, yielding the bimetallic Pd/Pt complex **80** in a 24% yield (Scheme 24). The analogous Pd/Pd bis-NHC complex could be obtained in the reaction of **78** with [Pd(PPh₃)₄] under somewhat milder conditions (18 h, 75 °C) in a 65% yield, showing the higher reactivity of Pd(0) compared to Pt(0) in oxidative additions.

The doubly chlorosubstituted bis-NHC precursor **78** served as the starting material for mono- and heterobimetallic NHC complexes via another synthetic route, too [45]. Although, in general, electron-rich tertiary phosphines are known to dehalogenate chloroimidazolium salts, the sterically encumbered tris(1,3-*tert*-butylimidazolidin-2-ylidenamino)phosphine (Scheme 25) was found to be particularly useful for the selective removal of chloride from 2-chloroazolium salts. Consequently, it was possible to react the linked 2-chlorobenzimidazole/2-chloro-3-ethylbenzimidazolium salt first with the mentioned phosphine, resulting in the exclusive metalation of the N-ethyl-benzimidazolium part, and then to perform oxidative addition at the chlorobenzimidazole end (Scheme 25). This strategy led to the isolation of an Ir/Pd heterobimetallic complex, **81**, but it has more general utility for obtaining mixed-metal NHC complexes.

**Scheme 24.** Regioselectivity of oxidative addition of C-Cl in **78** across Pd (0) in [Pd(PPh$_3$)$_4$] and Pt(0) in[Pt(PPh$_3$)$_4$], respectively.

**Scheme 25.** Synthesis of the Pd/Ir heterobimetallic NHC complex **81** via selective sequential reactions of C-Cl moieties in **78**, assisted by tris(1,3-*tert*-butilimidazolidin-2-ylideneamino)phosphine.

In the bis-NHC precursor **82**, an N-ethylimidazolium group and an 8-iodotheophylline unit were joined by an *m*-xylyl linker. This NHC precursor was cleanly monometalated at the imidazolium part with [IrCl$_2$(Cp*)]$_2$, and the resulting cyclometalated Ir(III) complex **83** was transformed into the Ir/Pd heterobimetallic bis-NHC product **84** under mild conditions (25 °C, in 1 day) (Scheme 26). The Ru/Pd heterobimetallic compound **85** was similarly obtained by the metalation of the imidazolium part of the propyl-bridged precursor with Ag$_2$O/[RuCl$_2$(η$^6$-*p*-cymene)]$_2$ (*p*-cymene = 4-isopropyltoluene), followed by the oxidative addition of C(8)-I in the 8-iodotheophylline unit to [Pd(PPh$_3$)$_4$] (Scheme 26) [46]. The coordinated carbonato ligand was formed by the oxidation of the CH$_2$Cl$_2$ solvent in the presence of silver salts.

**82** → **83** → **84**

**85**

**Scheme 26.** Synthesis of the Pd/Ir heterobimetallic NHC complex **84** by base-assisted deprotonation/Ir-coordination of the imidazolium moiety followed by oxidative addition of C-I bond in the 8-iodoteophylline unit across Pd(0). The same synthetic strategy allowed synthesis of the Pd/Ru heterobimetallic NHC complex **85**, too.

In another study [47], 4-bromo-3-methylthiazolium tetrafluoroborate was first metalated with $[IrBr_2(Cp^*)]_2$ with the use of the $Ag_2O$ method. This reaction yielded a mononuclear Ir-NHC complex, **86**, in which Ir was bound to the carbenic C2 atom of the heterocycle (Scheme 27). The subsequent oxidative addition of the C4-Br bond to Pd(0) in $[Pd(PPh_3)_4]$ at 90 °C in 24 h resulted in the heterobimetallic Ir/Pd complex **87**. Note that the overall reaction included a $Br^-/PPh_3$ exchange between the Ir and Pd centers. Under similar conditions, but at a temperature of 110 °C, the reaction of $[Pt(PPh_3)_4]$ afforded the analogous Ir/Pt heterobimetallic complex **88**, also with a $Br^-/PPh_3$ exchange between the two metal centers. The reaction of the substitutionally inert $[Pt(PPh_3)_4]$ for only 4 h allowed the isolation of the direct product of oxidative addition, an intermediate en route to **88**, still featuring a C4-carbene-bound $\{PtBr(PPh_3)_2\}$ moiety.

## 2.2.4. Pendant Group Coordination in the Synthesis of Heterobimetallic NHC Complexes and Their Catalytic Properties

N-heterocyclic carbenes containing a pendant donor group may show altered coordination (and catalytic) properties compared to their counterparts with no such substituents. Such pendant groups may serve as an anchor to facilitate the coordination of the N-heterocyclic carbene ligand, stabilize certain steric coordination geometries, and alter the electronic properties of the metal center in the carbene complex compared to complexes with exclusively NHC coordination. A case in point is the formation and catalytic properties of complexes obtained with the use of the NHC precursor **89** (Scheme 28), which was studied in much detail by Kühn, Baratta, and co-workers.

**Scheme 27.** Synthesis of the Pd/Ir (**87**) and Pt/Ir (**88**) heterobimetallic complexes from 4-bromo-3-methylthiazolium precursor by selective $Ag_2O$-assisted metalation with $[IrBr_2(Cp^*)]_2$ followed by oxidative addition of the C-Br bond across Pd(0) or Pt(0).

**Scheme 28.** Synthesis of the heterobimetallic NHC complexes **92**, **93** and **94** with the use of NHC ligand precursor **89** carrying a pendent -$PPh_2$ donor group.

The reaction of $[Ru(OAc)_2(PPh_3)_3]$ with one equivalent of **89**.HBr in the presence of NaOAc in *tert*-BuOH led to the clean formation of the mononuclear normal Ru(II)-NHC complex **90**. However, a second equivalent of **89** yielded a cationic Ru(II)-NHC product **91**, in which the second NHC ligand is coordinated on its C4-position (abnormal NHC) [48]. The formation of such a normal/abnormal NHC-coordinated metal complex may be due to the steric requirements of the bulky mesityl groups in **89**, together with the strong coordination of the substituent –$PPh_2$ group to Ru(II). The treatment of **91** with $Ag_2O$ afforded the cationic dimer **92** ($[AgBr_2]^-$ as a counterion), with the two **91** units connected by the bis-NHC coordination of Ag(I), which—at the same time—transformed the originally abnormal NHC moieties into N-heterocyclic dicarbene (NHDC) ligands (Scheme 28). Compound **92** was the first example of an NHDC complex containing two different d-block elements.

Compound **92** proved to be a versatile starting material for the preparation of heterobimetallic NHDC complexes. The reaction of this dimer with [AuCl(THT)] gave the

monomeric neutral Ru/Au compound **93**, while with [PdCl$_2$(cod)], the neutral Pd/Ru dimer **94** was formed, with a structure strictly analogous to that of **92** [49].

The monometallic **91** displayed very high catalytic activity in the transfer hydrogenation of various ketones, such as acetophenone, benzophenone, and cyclohexanone, to the respective alcohols. The reactions took place with a 0.1% catalyst loading in refluxing 2-propanol with NaO$^i$Pr as a base; turnover frequencies up to 49,000 h$^{-1}$ were achieved.

Catalyst **94** was studied in the tandem Suzuki–Miyaura *C–C* cross-coupling of various bromoacetophenones and boronic acids/transfer hydrogenation of the resulting ketones (Scheme 6a). In highly basic 2-propanol solutions, Suzuki–Miyaura cross-coupling was accompanied by extensive dehalogenation. However, under optimized conditions in toluene/2-propanol as a solvent and KOH as a base, yields of the respective alcohols in the range of 93–98% were obtained (according to GC or NMR analysis), showing the high functional group tolerance of the catalyst. In the reaction of 4-bromoacetophenone and phenylboronic acid, the time course of the reaction displayed the fast formation of the *C–C*-coupled product (acetylbiphenyl), followed by a slower transfer hydrogenation to 1-(4-biphenyl)ethanol. Under the same conditions, an equimolar mixture of **91** + [PdCl$_2$(IMes)$_2$] produced a lower yield of 1-(4-biphenyl)ethanol (76% vs. 95 with **94**) and the formation of several unidentified byproducts, thereby demonstrating the superior selectivity of the heterobimetallic catalyst **94** in this tandem Suzuki–Miyaura coupling/transfer hydrogenation.

The Ag(I)-linked cationic dimer **92** was applied to the synthesis of the neutral Ru/Ir heterobimetallic complex **95**, too, through its reaction with [IrCl(cod)]$_2$ via the Ag$_2$O route in CH$_2$Cl$_2$ (Scheme 29) [50]. With the use of cyclic voltammetry measurements, it was found that the Ru(II)/Ru(III) redox process occurred at a 70 mV lower potential in **95** than in **92**, suggesting an interaction between the Ir and Ru centers. For comparison, the normal/abnormal (NHDC) dimetalated (Ir/Ru) doubly phosphine-coordinated complex **96** was also prepared (Scheme 29). The solid-state structures of all new compounds were determined by X-ray crystallography.

**Scheme 29.** Synthesis of the heterobimetallic Ru/Ir NHC/phosphine complexes **95** and **96** containing normal/abnormal coordinated imidazole-diylidene ligands.

The catalytic activities of **91**, **95**, and **96** in the transfer hydrogenation of acetophenone were compared and are displayed in Table 1.

**Table 1.** Turnover frequencies of mono- and heterobimetallic Ru(II)-NHC complexes in transfer hydrogenation of acetophenone to 1-phenylethanol.

| Catalyst | Metal Center(s) | TOF (h$^{-1}$) |
|:---:|:---:|:---:|
| 91 | Ru (monometallic) | 26,000 |
| 95 | Ru/Ir (di-NHC) | 6700 |
| 96 | Ru/Ir (NHDC) | 1300 |

*Conditions*: Catalyst 0.1 mol%; NaO$^i$Pr (2 mol%); solvent: 2-propanol (5 mL); reflux temperature. Conversions were determined by GC; TOFs were calculated at 50% conversions.

It could be concluded from the data in Table 1 that the coordination of a second metal (Ir) decreased the catalytic activity of the mononuclear normal/abnormal Ru(II) NHDC complex to a large extent. Similar to the results of cyclic voltammetry, this also indicates a

strong metal–metal interaction; however, in this case, the interaction manifests itself in a large drop in catalytic activity. These data indirectly suggest that the N-C*H*-N imidazolium proton in **91** could play an important role in catalysis.

Kühn and co-workers metalated **91** with the use of Ag$_2$O and then [RhCl(cod)$_2$] to afford the Ru/Rh heterobimetallic NHDC complex **97** (Scheme 30) (solid-state structure determined by X-ray diffraction) with the intermediate formation of the Ag-linked dimer **92** [51]. The catalytic efficiency of **97** was tested in the transfer hydrogenation of acetophenone with basic 2-propanol as a H-donor (0.1 mol% catalyst, 2 mol% NaO$^i$Pr, 80 °C). In 25 min, the reaction reached equilibrium, which allowed the calculation of a TOF = 5700 h$^{-1}$. Under comparable conditions, the activity of the Ru/Ir complex **95** could be characterized by TOF = 6700 h$^{-1}$ (Table 1). A comparison of these data shows that, similar to the case of **95**, a significant drop in catalytic activity occurred upon the coordination of a second metal (Rh) to the mononuclear normal/abnormal bis(imidazolylidene)Ru(II) complex. However, the loss of catalytic activity was more or less the same in the case of the two complexes **95** and **97**, which may indicate the decisive role of the Ru(II) center in the studied complexes in the catalysis of acetophenone transfer hydrogenation.

**Scheme 30.** Synthesis of the Rh(I)-analog (**97**) of **95**.

*2.3. Synthesis and Catalytic Application of Heterobimetallic NHC Complexes Containing Triazolium-Derived Carbene Ligands*

1,2,4-Trimethyltriazolium salts can be deprotonated in methanol with the use of NaOMe, leading to the methanol adduct of the resulting monoylidene, which in turn can be metalated with a suitable metal compound, e.g., with [IrCl(cod)]$_2$, to yield the triazolium-monoylidene complex **98**. Further deprotonation and reaction with a metal complex provides—depending on the metal precursors used in the two steps—a homo- or heterobimetallic, mononuclear, or dinuclear complex of the 1,2,4-trimethyltriazole-diylidene (*ditz*) ligand **99** (Scheme 31). N-heterocyclic dicarbene (NHDC) complexes, formed from imidazole-based diylidenes (discussed in Section 2.2), are different from the *ditz*-based dinuclear complexes in that NHDC complexes are inherently asymmetric, while *ditz* derivatives may be symmetric when formed with the same metal fragments and N-substituents. What is common, though, in complexes of the two ligand sets is that the rigid NHC frame keeps the two metal ions at a given distance (usually close to 6 Å), which does not vary much with the nature of the metal ion and the *ditz* substituents. In other words, these complexes have no flexibility and, therefore, are unable to form metal chelates (in the absence of other donor atoms carried by the N-substituents).

**Scheme 31.** 1,2,4-Triazolium salts as precursors to heterobimetallic 1,2,4-trimethyltriazole-diylidene (*ditz*) complexes **99**.

The synthesis of heterobimetallic *ditz* complexes and their catalytic properties were reviewed by Mata, Hahn, and Peris [16], while Ventura-Espinoza and Mata [17] discussed the complex formation of *ditz* ligands containing other donor atoms, too. The formation of heterobimetallic complexes with 1,2,3-trialkyltriazolium salts as ligand precursors is also known [18]; however, their catalytic properties were less investigated.

The first synthesis of a heterobimetallic complex of the 1,2,4-trimethyltriazole-3,5-diylidene ligand was described by Mas-Marzá, Mata, and Peris in 2007 [52]. In the presence of KO$^t$Bu, [Ir(μ-Cl)(cod)]$_2$ was added to a THF solution of 1,2,4-trimethyltriazolium tetrafluoroborate at −40 °C, followed by the addition of more KO$^t$Bu and [Rh(μ-Cl)(cod)]$_2$ at room temperature, yielding the Ir/Rh heterobimetallic complex **100** in a moderate yield (Scheme 32). The reaction probably proceeded with the formation of a monometalated iridium species (not isolated). The analogous Ir/Ir and Rh/Rh homobimetallic complexes were also prepared (in these cases, in refluxing THF). These homobimetallic complexes proved to be excellent catalysts of the transfer hydrogenation of ketones and imines in basic (KOH) 2-propanol and also actively catalyzed the cyclization of 4-pentynoic and 5-hexynoic acids to the corresponding lactones. However, the catalytic properties of the Ir/Rh heterobimetallic compound **100** were not described in this early paper.

**Scheme 32.** Synthesis of the Ir/Rh heterobimetallic *ditz* complex **100** by sequential metalation.

A similar synthesis was worked out by Peris and co-workers with the use of the *ditz* precursor 1,2,4-trimethyltriazolium tetrafluoroborate and [Rh(μ-Cl)(cod)]$_2$, [Ir(μ-Cl)(cod)]$_2$, or [IrCl$_2$(Cp*)]$_2$ as metalation agents [53]. In these cases, the reactions proceeded in methanol, and NaH was used for the deprotonation of the triazolium salt. The first metalation step with [IrCl$_2$(Cp*)]$_2$ yielded the monometalated Ir(III) complex **101**. The addition of another batch of NaH and [Rh(μ-Cl)(cod)]$_2$ led to the formation of the heterobimetallic Rh(I)/Ir(III) *ditz* complex **102** (Scheme 33). In similar syntheses, with proper combinations of the Rh(I), Ir(I), and Ir(III) precursors, the analogous homobimetallic Rh(I)/Rh(I), Ir(I)/Ir(I), Ir(I)/Ir(III), and Ir(III)/Ir(III) *ditz* complexes were also obtained. The catalytic activity of these complexes was studied in the tandem cyclization/alkylation of 2-(*ortho*-aminophenyl) ethanol with benzyl alcohol and several other primary alcohols (Scheme 33). While in the reaction with benzyl alcohol, the use of the homobimetallic Ir(I)/Ir(I) and Ir(III)/Ir(III) complexes as the catalyst led to >95% conversion of the amino alcohol in 7 h, **102** showed inferior activity, with only 30% conversion in 20 h. This example shows that the connection of two different metal centers through a *ditz*-type ligand is not always advantageous but may also result in the diminished catalytic activity of the heterobimetallic complex relative to the homobimetallic analogs.

Various Ir/Pd bimetallic 1,2,4-triazole-diylidene complexes were prepared in a reaction of **101** with Pd(OAc)$_2$ in CH$_3$CN (**103**, **104**) and PdCl$_2$ in pyridine (**105**) (Scheme 34) [54]. These complexes were applied as catalysts in the tandem dehalogenation/transfer hydrogenation of 4-haloacetophenones (Scheme 34). The reaction of 4-bromoacetophenone catalyzed by the heterobimetallic **105** afforded 1-phenylethanol with 100% selectivity at >99% conversion (2 mol% catalyst, 2-propanol/Cs$_2$CO$_3$, 100 °C, 20 h, aerobic conditions). Under the same conditions, **103** and **104** were less effective, with 1-phenylethanol yields of 75% and 95%, respectively. Importantly, the mixture of the two homobimetallic complexes [{IrCl$_2$(Cp*)}$_2$(μ-*ditz*)] and [{PdCl$_2$(py)}$_2$(μ-*ditz*)] (*ditz* = 1,2,4-trimethyltriazole-3,5-diylidene; py = pyridine) showed high catalytic activity for the debromination of 4-bromoacetophenone but lower activity for transfer hydrogenation (72% acetophenone

and 25% 1-phenylethanol). Kinetic measurements revealed that with **105** as the catalyst, the reaction produced acetophenone and 4-bromophenylethanol as intermediates, which were subsequently hydrogenated and debrominated, respectively, resulting in the selective formation of 1-phenylethanol.

R = Pr, *i*Pr, Ph, *p*Cl-Ph

**Scheme 33.** Synthesis of the Ir/Rh heterobimetallic *ditz* complex **102** and its use as catalyst in tandem cyclization/alkylation of 2-(*ortho*-aminophenyl) ethanol with primary alcohols.

**Scheme 34.** Synthesis of the Ir/Pd heterobimetallic (**103**, **105**) and heterotrimetallic (**104**) *ditz* complexes and their use as catalysts in tandem Suzuki-Miyaura coupling/transfer hydrogenation/alkylation of 4-bromoacetophenone in 2-propanol (**a**) and in *n*-butanol (**b**).

Compound **105** also catalyzed the Suzuki–Miyaura *C*–*C* cross-coupling of 4-bromoaceto-phenone and phenylboronic acid (Scheme 34) in the presence of alcohols. The reaction proceeded with simultaneous debromination, which first yielded 4-phenylacetophenone. When a short reaction time (0.5 h) was applied in the presence of benzyl alcohol, *n*-butanol, and *t*-butanol, the yield of this intermediate was determined by GC analysis to be in the range of 93–96%. The further reaction of this intermediate with the respective alcohols resulted mainly in the formation of the α-alkylated products, which were isolated in 69–86% yields after 20 h (2 mol% catalyst, 2 mL R-OH, 2 mL THF, 100 °C). Independent of the reaction time, the yield of 1-(4-biphenyl)ethanol arising from the transfer hydrogenation of 4-phenylacetophenone was significant (up to 88% in 7 h) only when 2-propanol was used in the solvent mixture. Again, the mixture of the two homobimetallic complexes [{IrCl$_2$(Cp*)}$_2$(μ-*ditz*)] and [{PdCl$_2$(py)}$_2$(μ-*ditz*)] was less effective in the α-alkylation step, affording only 29% of the α-butyl-4-phenylacetophenone. Together with the results on tan-dem Suzuki–Miyaura *C*–*C* cross-coupling, this observation clearly showed the advantage of using the two different metals in a well-defined Ir/Pd triazole–diylidene complex in comparison to the application of a mixture of analogous Ir/Ir and Pd/Pd complexes—a convincing example of metal–metal cooperativity within the **105** catalyst.

In a related study, Zanardi, Mata, and Peris obtained the Pt/Ir heterobimetallic com-plex **106** in the reaction of PtI$_2$ and **101** in pyridine using NaH as a deprotonating agent (Scheme 35) [55]. When PtI$_2$ reacted with trimethyltriazolium tetrafluoroborate in the presence of K$_2$CO$_3$ in pyridine, the homobimetallic compound [{PtCl$_2$)(py)}$_2$(μ-*ditz*)], **107**, was obtained. Both compounds were analyzed by single-crystal X-ray diffraction, and the metal–metal distances in the Pt/Ir and Pt/Pt complexes were found to be 6.04 Å and 5.96 Å, respectively. The catalytic properties of **106** and **107** were compared in the tandem oxidative cyclization of 2-(*ortho*-aminophenol)ethanol to indole (typically an Ir-catalyzed reaction), followed by the addition of alkynyl alcohols (generally a Pt-catalyzed process). In a typical reaction, the first step (oxidative cyclization to indole) was run for 12 h, followed by the addition of the alkynyl alcohol to provide the functionalized indole derivative (2 mol% catalyst, toluene, 110 °C, total reaction times 12–24 h) (Scheme 35; it shows only the second step of the tandem reaction). The yields of functionalized indoles (determined by GC analysis) with catalyst **106** were in the range of 74–81% (depending on the alkynyl alcohol used). In contrast, **107** alone did not catalyze the tandem process at all; however, an equimolar mixture of **107** and [{IrCl$_2$(Cp*)}$_2$(μ-*ditz*)] afforded the functionalized indole derivatives in up to 44% yield. Note that [{IrCl$_2$(Cp*)}$_2$(μ-*ditz*)] alone was more active (56% product yield) than its mixtures with **107**, and its activity was close to that of the heterobimetallic Ir/Pt complex **106** (74%). These studies led to the conclusion that some cooperation between the Ir and Pt centers in **106** may contribute to the favorable outcome of the tandem oxidative cyclization/addition reactions of 2-(*ortho*-aminophenol)ethanol and alkynyl alcohols, although this effect was not strongly manifested.

Zanardi, Mata, and Peris also investigated the one-pot preparation of imines from nitroarenes in a borrowing-hydrogen process (Scheme 36) catalyzed by complexes [IrCl$_2$(Cp*) ((μ-*ditz*)PdCl$_2$(py)] (**105**; Ir/Pd)), [{PdCl$_2$(py)}$_2$(μ-*ditz*)] (Pd/Pd), and [{IrCl$_2$(Cp*)}$_2$(μ-*ditz*)] (Ir/Ir) [56]. Condensation reactions of nitrobenzene and 4-methyl- and 4-methoxynitrobenzene with primary alcohols (benzyl, 4-methyl-, and 4-methoxybenzyl alcohol, 2-phenylethanol) were most efficiently catalyzed by **105** (Ir/Pd), with GC yields up to 92% (2 mol% catalyst, benzyl alcohol as solvent and reagent, Cs$_2$CO$_3$, 110 °C, 3–20 h, aerobic conditions). Equimo-lar mixtures of the above homobimetallic Pd/Pd and Ir/Ir complexes also showed high catalytic activity in the reaction, although with a few substrates, such as (4-methoxy)benzyl alcohol, their catalytic activity did not reach that of **105** (yields were 55% and 75%, respec-tively). Interestingly, in the reaction of 4-bromonitrobenzene and benzyl alcohol catalyzed by **105**, in the presence of phenylboronic acid, Suzuki–Miyaura *C*–*C* cross-coupling took place, too, yielding the corresponding bisarylated imine (Scheme 36). The analogous ho-mobimetallic Ir/Ir complex, [{IrCl$_2$(Cp*)}$_2$(μ-*ditz*)], did not afford the bisarylated imine

product at all, thus demonstrating the favorable effect of having two catalytically active metal centers within one well-defined molecule.

**Scheme 35.** Synthesis of the Ir/Pt heterobimetallic *ditz* complex **106** and its use as catalyst in alkyne addition to indoles.

**Scheme 36.** Condensation reactions of nitrobenzene with primary alcohols catalyzed by **105** (**a**) in the absence, and (**b**) in the presence of phenylboronic acid.

The *ditz* ligand was also used in the synthesis of the Ir/Au heterobimetallic complex **108** via the Ag$_2$O route from **101** and AuCl(SMe$_2$). Interestingly, the analogous *ditz*-bridged homobimetallic Au(I) complex could not be obtained; instead, 4-ethyl-1,2-dimethyltriazolium tetrafluoroborate had to be used, possibly for solubility reasons (Scheme 37) [57]. The catalytic efficiencies of the two complexes were compared in the reaction of various substituted nitrobenzenes and benzyl alcohol. With nitrobenzene and 4-methylnitrobenzene substrates, the heterobimetallic **108** catalyzed the formation of the corresponding imines (Scheme 36) with excellent yields in the range of 93–97%; however, 4-methoxynitrobenzene provided only a 41% imine yield. Nevertheless, in all cases, the yields were substantially higher than those obtained with the close analog Au/Au complex [(AuCl)$_2$(μ-4-ethyl-1,2-triazolediylidene)]. Compound **108** effectively catalyzed the reaction of azoxybenzene and benzyl alcohol, leading to the formation of azobenzene; however, azobenzene could not be further reduced to the corresponding imine (or amine). This allowed the conclusion that the imine (and the amine) products were formed by the direct condensation of nitrobenzene and benzyl alcohol.

**Scheme 37.** Synthesis of the Ir/Au heterobimetallic *ditz* complex **108** and its use as catalyst in reaction of various substituted nitrobenzenes and benzyl alcohol.

The heterobimetallic Ir/Ru complex, **109**, containing the *ditz* dicarbene ligand, was obtained from **101** and [RuCl₂(*p*-Cym)]₂ (*p*-Cym = *p*-cymene, 4-isopropyltoluene) in acetone in the presence of NaOAc at 50 °C [58]. This complex could also be prepared directly from 1,2,4-triazolium tetrafluoroborate by deprotonation with NaH in methanol, followed by the addition of [RuCl₂(*p*-Cym)]₂, and a subsequent reaction of the isolated monometalated Ru(II)-triazolylylidene complex **110** [58] with [IrCl₂(Cp*)]₂ in acetone/NaOAc (Scheme 38). Compound **109** was tested as a catalyst for the chelation-assisted arylation of 2-phenylpyridine with chlorobenzene in the Oppenauer oxidation of 1-phenylethanol and in the unprecedented tandem process of the oxidation of 1-(4-halophenyl)ethanols and the arylation of arylpyridines. In the latter reaction (Scheme 38), the use of **109** led to high yields of bis(2,6-diarylated)arylpyridines (76% with 2-phenylpyridine); however, this activity was inferior to that of other related monometalated *ditz* and imidazolylylidene Ru-carbenes. It seems that in **109**, the Ru and Ir ions act as independent catalytic centers, and their weak electronic interaction (determined by cyclic voltammetry and differential pulsed voltammetry) results in some loss of the overall catalytic activity of this heterobimetallic complex in comparison to other homometallic and mono- or binuclear Ru(II)-NHC complexes used in this study.

The reaction of **101** with chiral *N,N*-dimethylbenzylaminate palladacycles afforded the chiral at palladium heterobimetallic (Ir/Pd*) complexes **111**, **112**, **113**, and **114** (Scheme 39) [59]. X-ray-quality single crystals of **111** could be isolated by the slow evaporation of dichloromethane/hexane mixtures. The crystal structure contained two of the four possible atropisomers, the same two that could also be identified in solution by NMR spectroscopy. The four compounds were used as catalysts for the tandem isomerization/asymmetric hydrophosphination of 1,3-diphenylpropargyl alcohol (Table 2). In all cases, isomerizations were performed for 5 h at 90 °C, while hydrophosphinations proceeded at room temperature for 10 h, both with 2 mol% Ir/Pd* and 2-propanol/K₂CO₃ (these data are not shown in Table 2). All four complexes were excellent catalysts for the isomerization of the substrate to chalcone (GC yields of 90–97%). In the second step, the hydrophosphination of the latter with HPPh₂ afforded products with a C-3 phosphorus addition in good-to-moderate isolated yields (56–75%) and excellent regioselectivity. Unfortunately, only low enantioselectivities (4–17% *ee*) were reached in this tandem reaction, according to chiral HPLC determinations using the oxides of the hydrophosphination products. From these data, it could be concluded that the isomerization step was catalyzed mostly by the Ir fragment of the catalysts (this is why all isomerizations were fast and practically quantitative), while the Pd* fragment

was responsible for hydrophosphination. The low enantioselectivity (similarly observed in hydrophosphinations with Pd-NHC catalysts) [60] was attributed to the small size of *N*-substituents (methyl groups) in the Pd* fragment of the catalysts.

**Scheme 38.** Synthesis of the Ir/Ru heterobimetallic *ditz* complex **109** and its use as catalyst in (**a**) arylation of 2-phenylpyridine, (**b**) Oppenauer oxidation of 1-phenylethanol and (**c**) tandem oxidation of 1-(4-halophenyl)ethanols/arylation of arylpyridines.

**Scheme 39.** Synthesis of chiral at palladium (Ir/Pd*) heterobimetallic *ditz* complexes **111–114** in reaction of **101** with *N,N*-dimethylbenzylaminate palladacycles.

**Table 2.** Sequential catalytic isomerization/hydrophosphination of 1,3-diphenylpropargyl alcohol.

| Catalyst | Isomerization | Hydrophosphination | |
|---|---|---|---|
| | Yield [%] (GC) | Yield [%] (Isolated) | *ee* [%] |
| **111** | 95 | 60 | 4 |
| **112** | 97 | 75 | 8 |
| **113** | 95 | 71 | 14 |
| **114** | 90 | 56 | 17 |

　　Sabater, Mata, and Peris synthesized the heterobimetallic Ru/Pd complex **115** in a reaction of **110** and the acetate-bridged *N,N*-dimethylbenzylaminate palladacycle dimer in an acetone solution in the presence of NaCl at 50 °C [61]. The reaction of the same palladacyle and 1,2,4-trimethyltriazolium tetrafluoroborate yielded the related homobimetallic Pd/Pd complex **116**, with the *ditz* ligand linking the two palladacyle fragments (Scheme 40). Compounds **115** and **116** and the earlier synthesized [{RuCl$_2$(*p*-Cym)}$_2$(μ-*ditz*)] **117** [62] were applied as catalysts for the hydrodefluorination of aromatic and aliphatic carbon-fluorinated compounds. Due to the strength of the C–F sigma bonds, the design of practically applicable catalysts for defluorination under mild conditions is a major challenge. It was found by Peris and co-workers that 1 mol% **115** (Ru/Pd) actively catalyzed the defluorination of fluorobenzene to benzene, reaching 100% conversion in 1 h at 80 °C in the presence of an equimolar amount of KO$^t$Bu in 2-propanol as the solvent and hydrogen donor (Scheme 40). This high activity is in striking contrast to the inefficiency of **117** (Ru/Ru) and **116** (Pd/Pd) (0 and 3% yields of benzene, respectively). Applying a mixture of 0.5 mol% **117** and 0.5 mol% **116** resulted in only a 49% yield. These observations showed that efficient defluorination required both metal ions; however, having them built into the same well-defined Ru/Pd molecule (such as **115**) offered profound advantages. Compound **115** showed outstandingly high catalytic activity in the defluorination of a series of 11 monofluorinated aromatic substrates. In addition, *p*- and *o*-difluoro- and 1,3,5-trifluorobenzenes could be dehalogenated to benzene with a 100% yield. In all investigated cases, **117** proved to be inactive for defluorination, and **116,** as well as mixtures of **117** and **116**, were also inferior to **115** in the catalysis of hydrodefluorination. In addition, **115** provided a 50% yield of toluene in the transfer hydrodehalogenation of α,α,α-trifluorotoluene, admittedly one of the most inert organofluorine compounds (Scheme 40). Other (trifluoromethyl)arene compounds reacted similarly well, and in the case of 1-(trifluoromethyl)naphthalene, the isolated yield reached 99%. Importantly, no partially defluorinated intermediates were detected, even in reactions with incomplete conversion.

　　Maity and co-workers synthesized a bis(1,2,3-triazolyl)naphthalene derivative in a copper-catalyzed click reaction of 1,4-bis(ethynyl)naphthalene with in situ-generated ethyl azide. The product was converted to the corresponding bis-triazolium salt using methyl iodide. The bis-triazolium salt could be deprotonated in a stepwise manner, such that its reaction with [IrCl$_2$(Cp*)]$_2$ in the presence of Cs$_2$CO$_3$, NaOAc, and KI in acetonitrile led to selective monometalation, yielding the cyclometalated Ir-NHC complex **118** (Scheme 41). The subsequent metalation of the other triazolium fragment of the molecule with PdCl$_2$ in pyridine in the presence of K$_2$CO$_3$ and KI provided the heterobimetallic (Ir/Pd) bis(triazolylidene) complex **119** [63]. The analogous homobimetallic Pd/Pd complex **120** could be obtained in one step from the bis-triazolium salt. The catalytic properties of **119** were investigated in the tandem Suzuki–Miyaura *C–C* cross-coupling/transfer hydrogenation of 4-bromoacetophenone (and, to a lesser extent, 3-bromoacetophenone) in basic 2-propanol-THF solutions at 90 °C. Compound **119** itself showed only modest catalytic activity for the production of the respective 1-biphenylethanols; however, the addition of PPh$_3$ increased the yield from 16% to 65%. Catalytic Sonogashira *C–C* cross-coupling

was actively catalyzed by **119** in the presence of PPh$_3$ (yield of the biaryl product was 75% with 4-bromacetophenone and 90% with 3-bromoanisole, respectively), and the yields were only slightly increased upon the addition of CuI as a co-catalyst. It is of interest that the yield of the reaction of 4-bromoacetophenone was slightly lower with the heterobimetallic **119** + PPh$_3$ than with the homobimetallic **120** + PPh$_3$ catalysts (75% vs. 80%). Catalyst **119** was also applied in the reaction of oxindole with 2-bromotoluene, but without an additive, it showed negligible catalytic activity. Again, the addition of PPh$_3$ resulted in a better yield (51% with 6 mol% catalyst in toluene at 120 °C for 20 h); however, this catalytic activity was still considerably lower than that of the homobimetallic **120** + PPh$_3$ (72% with 3 mol% catalyst, the same conditions as for the reaction catalyzed by **119** + PPh$_3$). Consequently, in this case, no favorable effect of the use of a heterobimetallic bis-triazolylidene complex over the analogous homobimetallic catalyst was observed, which may be because of the relatively large Ir-Pd distance, 8.947 Å, in **119**.

**Scheme 40.** Synthesis of the Ru/Pd heterobimetallic *ditz* complex **115** and its use as catalyst in (**a**) transfer hydrodefluorination of fluorobenzenes to benzene derivatives and (**b**) transfer hydrodefluorination of α,α,α-trifluorotoluene to toluene.

*2.4. Synthesis and Catalytic Application of Heterobimetallic NHC Complexes Containing Linked Imidazolium- and/or Triazolium-Derived Carbene Ligands*

Zamora, Ferguson, and Cowie synthesized a 1,3-phenylene-linked bis-triazole in the click reaction of 1,3-dialkynylbenzene and 2-azido-1,3-diisopropylbenzene, which was converted to the linked bis-triazolium trifluoroacetate **121** with the use of methyl triflate in CH$_2$Cl$_2$ at −78 °C [64]. Compound **121** could be stepwise deprotonated and metalated simultaneously with the use of [M(μ-OMe)(cod)]$_2$ (M = Ir, Rh), reactions in which the methoxy group plays the role of an internal base. For example, the first reaction with [Ir(μ-OMe)(cod)]$_2$ yielded the half-metalated **122** Ir-triazolilydene complex, still having a pendant triazolium moiety. The reaction of the latter with [Rh(μ-OMe)(cod)]$_2$ led to the formation of the heterobimetallic Ir/Rh bis-mesoionic carbene (MIC-MIC) complex **123** (Scheme 42). All new compounds were thoroughly characterized by NMR spectroscopy; however, no X-ray diffraction study of **123** could be performed in in lack of suitable single crystals. Furthermore, the catalytic properties of the new complexes were not investigated.

In a related study, Cowie and co-workers synthesized unsymmetrical imidazolium/triazolium dicarbene precursors, which were subsequently used for the preparation of homo- and heterobimetallic NHC/MIC complexes [65]. The synthesis consisted of a click reaction of 1-alkyl-3-alkynyl imidazolium bromides with azidomethylbenzene or 2-azido-1,3-diisopropylbenzene. The product was converted to the bis-azolium salt **124** with methyl triflate. The stepwise metalation of **124**, first with Pd(OAc)$_2$ and then with [Rh(μ-OMe)(cod)]$_2$ (both steps in the presence of KI with heating in acetonitrile), resulted in the heterobimetallic NHC/MIC Pd/Rh complex **125** (Scheme 43). All new complexes

were thoroughly characterized; however, repeated attempts to obtain single crystals of **125** for X-ray measurements failed. No catalytic studies were described in this paper.

**a**

**b**

**c**

**Scheme 41.** Synthesis of the heterobimetallic Ir/Pd bis(triazolylidene) complex **119** and its use as catalyst in (**a**) Suzuki-Miyaura *C–C* coupling of aromatic aldehydes and phenylboronic acids, (**b**) Sonogashira *C–C* coupling of phenylacetylene and bromoarenes, (**c**) arylation of oxindole with 2-bromotoluene.

Dipp = 1,3-diisopropylphenyl

**Scheme 42.** Synthesis of the heterobimetallic Ir/Rh complex **123** containing linked triazolylidene units in the ligand.

Dipp = 1,3-diisopropylphenyl

**Scheme 43.** Synthesis of the heterobimetallic Pd/Rh complex **125** containing linked imidazolylidene/triazolylidene units in the ligand.

As part of a more general approach, Mendoza-Espinoza et al. synthesized the methylene-linked imidazolium-triazolium dicationic bis-carbene precursor **126** in a copper-catalyzed click reaction of 1-mesityl-3-alkynylimidazole and mesityl azide. The deprotonation of **126** with slightly less than equimolar KHMDS (K-hexamethyldisilazane), followed by the addition of [PdCl₂(allyl)]₂ or [Rh(μ-Cl)(cod)]₂, resulted in Pd or Rh metalation, respectively, on the imidazole C2, leaving the triazolium part available for subsequent reactions. Indeed, the further reaction of this intermediate with [PdCl₂(allyl)]₂ or [Rh(μ-Cl)(cod)]₂ supplied the bimetallic NHC/MIC complex **127** or **128**, in which the bonding of the metal ions to NHC carbene or 1,2,3-triazolyl carbene atoms depended on the order of addition of the Pd and Rh complexes (Scheme 44) [66]. All complexes were fully characterized by NMR methods; however, attempts to obtain single crystals of **127** and **128** were unsuccessful. Possible catalytic uses of the heterobimetallic NHC/MIC complexes were mentioned in the paper; however, no such reactions were described.

**Scheme 44.** Synthesis of the heterobimetallic Pd/Rh complexes **127** and **128** containing linked imidazolylidene/triazolylidene units in the ligand.

In a recent article, Longhi et al. discussed possible synthetic approaches for heterobimetallic complexes by using imidazolylidene/triazolylidene heteroditopic carbene ligands [67]. Their contribution to the field was the synthesis of a propyl-linked imidazole/1,2,3-triazolium compound **129** in the reaction of a N-substituted imidazole and 5-chloro-1-pentyne, followed by a copper-catalyzed click reaction of this intermediate with phenylazide. The resulting compound was methylated at the N3 position of the triazole moiety, giving the pro-ligand **130** (similar to **126**) (Scheme 45).

**Scheme 45.** Synthesis of the heterobimetallic Ag/Au complex **131** containing propyl-linked imidazolylidene/triazolylidene ligands in bridging position.

The heterobimetallic complex **131** was not prepared directly from **130**. Instead, the bis-NHC-Au complex of **129** was methylated with the use of Me₃OBF₄ on the triazole ring. The further reaction of the product with Ag₂O yielded the heterobimetallic Au/Ag complex **131**, in which Au acts as a bride to the imidazolyl carbene atoms, while Ag forms a bridge between the triazolilydene carbene donor atoms (Scheme 45). No catalysis with this complex was reported.

## 3. Conclusions and Perspectives

The idea of using more than one different metal bound to the same ligand entity (monomeric, oligomeric, or polymeric) has long fascinated scientists working on homogeneous metal-complex catalysis. In addition to speeding up sequential (consecutive) reactions, resulting in higher yields, the cooperation of different metals in close proximity to each other may open new pathways for synthetic reactions, yielding new useful products.

This review treats only those complexes in which at least two different metals are bound to the same ligand by at least one M-C(carbene) bond each. Still, it convincingly demonstrates the developments of the chemistry of heterobimetallic complexes of N-heterocyclic (including mesoionic) carbene ligands. Such ligands included linked imidazolium (NHC) derivatives, normal/abnormal NHCs, and 1,2,4- or 1,2,3-triazolylidenes (or their combinations). Various ingenious methods have been developed for the synthesis of such complexes, most of all by the stepwise classical deprotonation/metalation of the ligand. In several cases, deprotonation/metalation with one metal (M) was followed by the oxidative addition of a C(carbene)–halogen bond onto a low-valent metal ion (M′), such as in [Pd(PPh$_3$)$_4$].

So far, the studied catalytic processes covered only a relatively narrow field of tandem or consecutive synthetic transformations, such as the tandem dehalogenation/transfer hydrogenation of bromoacetophenones in 2-propanol solutions or the Suzuki–Miyaura *C–C* coupling of brominated aromatic ketones or aldehydes, followed by transfer hydrogenation (in 2-PrOH) or α-alkylation (in *n*BuOH). Perhaps the most fascinating finding was the easy defluorination of fluorobenzol (and its derivatives) and α,α,α-trifluorotoluene catalyzed by a Ru/Pd heterobimetallic complex (**115**) of the 1,2,4-trimethyltriazole-diylidene (*ditz*) ligand under mild conditions. These reactions clearly show the power of metal–metal cooperativity in a heterobimetallic dicarbene complex, for example, in breaking up aromatic and aliphatic C-F bonds, among the strongest ones in organic chemistry. The relatively low number of catalytic reactions investigated with the use of bimetallic NHC catalysts is partly due to the requirement for the high stability of both M-C(carbene) and M′-C(carbene) bonds in the same molecule under the conditions required for the tandem processes. However, since a large number of bimetallic NHC complexes obtained so far have not been applied to catalysis, it can be expected that the number of synthetic procedures with such catalysts will increase in the future.

In several cases, heterobimetallic di- or polynuclear NHC complexes showed higher (sometimes significantly higher) catalytic activity than those of the analogous but monometallic complexes of the same ligand or than mixtures of reasonably comparable mononuclear building units of the heterobimetallic NHC-complex catalysts, which contained the different metals in separate 'fragments'. However, several examples were also found that showed that the addition of a second metal to another C(carbene) donor atom of the ligand did not improve, or even significantly decreased, the catalytic activity of the homometallic carbene complex obtained with only the first metal. These findings call attention to more detailed mechanistic investigations to reveal the underlying effects leading to increased or decreased catalytic activity. Targeted investigations are also needed of relationships between the catalyst structure and catalytic activity/selectivity, especially in those cases where the tandem reactions showed the unambiguous catalytic cooperativity of the different metal centers (and were not only the usual consecutive reactions catalyzed by the two metal centers independently).

The stability of the heterobimetallic complexes under conditions of catalysis (decomposition pathways, products, and kinetics) was rarely studied, except for the Hg test of homogeneity in a few cases. Knowledge of the fate of the catalytically active species during a catalytic run could largely aid the development of robust, active, and selective catalysts. Similarly, no attempts have been made so far to recover and reuse heterobimetallic catalysts that are either anchored onto a suitable solid support or in liquid–liquid biphasic reaction systems. Such studies would be advantageous for practical applications.

We are convinced that this most intriguing field of homogeneous metal-complex catalysis will bring further interesting and important results.

**Author Contributions:** Conceptualization, F.J. and C.E.C.; Methodology, G.P.; Writing—Original Draft Preparation, F.J.; Writing—Review and Editing, F.J., Á.K. and C.E.C.; Visualization, G.P.; Supervision, F.J.; Funding Acquisition, F.J. and G.P. All authors have read and agreed to the published version of the manuscript.

**Funding:** This research was supported by the EU and co-financed by the European Regional Development Fund under the projects GINOP-2.3.3-15-2016-00004 and GINOP 2.3.2-15-2016-00008. Financial support from project no. RRF-2.3.1-21-2022-00009, titled National Laboratory for Renewable Energy, has been provided by the Recovery and Resilience Facility of the European Union within the framework of Programme Széchenyi Plan Plus, which is gratefully acknowledged.

**Data Availability Statement:** Not applicable.

**Conflicts of Interest:** The authors declare no conflict of interest.

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
