# Peer review of "Heterobimetallic Complexes of Bi- or Polydentate N-Heterocyclic Carbene Ligands and Their Catalytic Properties"

_catalysts, doi:10.3390/catal13111417_

Round 1
Reviewer 1 Report
Comments and Suggestions for Authors
This seems to be a well constructed review on a focused topic. It is dense but well organized and the compiled material can be valuable to chemists in and outside the field. I have made a number of small corrections and suggestions in a PDF for the authors to check, but I think the manuscript can be recommended for publication after some minor corrections are applied.

Comments on the Quality of English LanguageThe English is fair and the prose can be read easily, no issues here.
Reviewer 2 Report
Comments and Suggestions for Authors
The review of Gábor Papp and co-authors is about heterobimetallic NHC-carbene complexes, their history, synthetic methods and catalytic properties. The review consists of 37 pages with 67 references. The review has a lot of good quality chem schemes which help to navigate and understand manuscript.
The author made a really great job. Not every review that gets reviewed is of such high quality. First of all, the meaning of the work is clear thanks to a very good introduction. Synthesis methods are described in detail, almost as in the original articles. This review could be a useful source of synthetic methods. It is also very respectful that the authors choose to print the review without pay-wall in open access journal so that as many people as possible can read the above.
I have almost no questions regarding the content of the work, except for some of the references used.
Reference 6, as far as I know, is a review article on the synthesis of heterometallic compounds, but not on " The application of catalysts…"
It is not clear why the authors, when providing historical information about the discovery of N-heterocyclic carbenes, refer to paper “W.A. Herrmann, N-Heterocyclic Carbenes: A New Concept in Organometallic Catalysis, Angew. Chem. Int. Ed. 41 (2002) 1290–1309.” while carbenes are much older, at least A. J. Arduengo III., R. L. Harlow, M. J. Kline: A stable crystalline carbene, J. of the Am. Chem. Soc. Band 113, 1991, 113, S. 361–363 https://doi.org/10.1021/ja00001a054
On page 25 and onwards: “1,2,4-Trimethyltriazolium salts can be deprotonated in methanol with the use of NaH” - There is probably an inaccuracy here. Most likely, it is need sodium methoxide. The hydride will react with methyl alcohol to form sodium methoxide? Yes, that's right, I checked reference 53 - it says the following: “Method b. A suspension of sodium hydride (60% in mineral oil, 17 mg, 0.42 mmol) in dry methanol was stirred at 0 °C under a nitrogen atmosphere until a clear solution of sodium methoxide was formed.” Therefore, clarity should be provided in the current review when writing the reaction scheme for sodium hydride in methanol.
In any case, the presented review is a very interesting work and I certainly recommend accepting it for publication in the journal Catalysts. This work will definitely attract a lot of attention from researchers and increase citations for the Catalysts.
